# On the difficult evolutionary transition from the free-living lifestyle to obligate symbiosis

**Phuong Linh Nguyen**[¤‡*], **Minus van Baalen**[‡]

Institut de Biologie de l'École Normale Supérieur, Paris, France

¤ Current address: Institut d'écologie et des sciences de l'environnement de Paris, Paris, France
‡ These authors also contributed equally to this work.
* lpnguyen@biologie.ens.fr

**Data Availability Statement:** The data underlying the results presented in the study are obtained from model simulation. The codes that generated

## Abstract

Obligate symbiosis evolved from free-living individuals most likely via the intermediate stage of facultative symbiosis. However, why should facultative symbionts, who can live independently but also benefit from their partners if these are available, give up this best of both worlds? Using the adaptive dynamics approach, we analyse a simple model, focusing on one partner of the symbiosis, to gain more insight into the selective forces that make individuals forgo the ability to reproduce in the free-living state. Our results suggest that, similar to the parasitism-mutualism continuum, the free-living way of life and obligate symbiosis are two extremes of a continuum of the ability to reproduce independently of a partner. More importantly, facultative symbiosis should be the rule as for many parameter combinations completely giving up independent reproduction or adopting a pure free-living strategy is not so easy. We also show that if host encounter comes at a cost, individuals that put more effort into increasing the chances to meet with their partners are more likely to give up the ability to reproduce independently. Finally, our model does not specify the ecological interactions between hosts and symbionts but we discuss briefly how the ecological nature of an interaction can influence the transition from facultative to obligate symbiosis.

## Introduction

Many ecological interactions involve prolonged and intimate contact between two or more individuals, a phenomenon called symbiosis that may involve both mutualism and parasitism [1]. Since symbiosis was identified as a potential for evolutionary innovation [2–6], the conditions that favour it and its subsequent evolutionary outcomes have received much attention. Studies include the evolution of organelles from symbiotic relationships [7, 8], the maintenance of mutualism [9–14], the evolution of virulence [8, 15–20], and the transition along the parasitism-mutualism continuum (theoretical work: [21–25], empirical work: [26–29]).

The interaction in symbiosis, whether it is parasitism or mutualism, is conventionally classified as obligate or facultative. Obligate symbiosis suggests that at least one of the partners cannot complete its lifecycle on its own while facultative symbiosis suggests that partners do not

the data and figures are included in two uploaded mathematica files.

**Funding:** This work has received support under the program « Investissements d'Avenir » launched by the French Government and implemented by ANR with the references ANR- 10-LABX-54 MEMOLIFE and ANR-11-IDEX-0001-02 PSL* Research University. The fund for the author Linh Phuong NGUYEN is under the "Contrat Doctoral no2016-2" by the Ecole Normale Superieur and PSL Research University. The sponsors do not play any role in the study design, data collection and analysis, decision to publish or preparation of the manuscript.

**Competing interests:** The authors have declared that no competing interests exist.

necessarily require one another to survive and/or reproduce. Moreover, this need for association may be asymmetric for the partners, that is, one partner may need the other but not necessarily vice versa, for instance, parasites may need hosts to complete their lifecycles but hosts definitely do better on their own. In mutualistic relationships, the assignment of 'host' and 'symbiont' is often arbitrary; and in many cases, the roles of host and symbiont are interchangeable. It is therefore important to specify which partner in an association the facultative and obligate relationship is referred to. In this article, we focus on the evolution of the dependency of only one partner, the symbiont, on the other partner, the host.

Obligate symbionts have to evolve from free-living individuals, regardless of whether they are parasites or mutualists, because if there were no free-living individuals to begin with, there would be no ingredients for associations. If this process occurs due to, for instance, an event of engulfment of the symbiont by the host, followed by purely vertical transmissions of the symbionts to maintain the population then obligate symbionts may evolve directly from free-living organisms. However, many symbionts have to experience external environment before encountering new hosts, and facultative symbiosis is much likely an intermediate step leading toward obligate symbiosis. From an adaptationist point of view, the evolutionary transition to obligate symbiosis presents an evolutionary riddle since a facultative symbiont always has the option to live autonomously without a host, so why give that up?

Indeed, facultative symbioses persist in many cases: among the parasites, for instance, Cholera pathogens, causing widespread epidemics, maintain a capacity for independent reproduction [30]. And among the more mutualistic interactions, many soil organisms that engage in cooperative interactions with plants can reproduce independently of their plant partners [31]. However, many times in evolutionary history, lineages have given up the capacity of independent reproduction, at least in nature [32]. The question of whether a given symbiont is obligate thus boils down to whether they are able to reproduce on their own, in absence of any potential host. This forces us to take the entire lifecycle of the species into account, which often is outside of the knowledge accessible by experiments. Our work, therefore, will be useful for both future theoretical and experimental studies.

The question of why facultative symbiosis can become obligate is similar to why generalists may give up the ability to exploit some resources and become specialists, but there are subtle differences too. Rueffler et al. [33] showed that evolutionary consequences changed when density dependence and different evolving life-history traits were taken into account. In generalist-specialist models, very often the ecological dynamics of the two resources are simple and their interactions with the consumer are symmetric. Thus, the trade-off often assumes a single trait that affects consumption such that consumption efficiency of one resource directly affects consumption efficiency of the other because the trait has equivalent performances in the two resources. For instance, big beaks are more efficient in cracking big seeds and small beaks are more sophisticated to open small seeds but birds have but one beak.

In contrast, host-symbiont interactions and external resource-symbiont interactions are typically asymmetric, hence, a trait that contributes to the former may not play the equivalent role in the latter or even any role at all. Trade-offs then may be among different traits. More importantly, the symbiont influences its host not only via its direct effects in the association but also through the dynamics of both unoccupied hosts and associations while, in contrast, the interactions between consumers and resources are often short, and the dynamics of such short associations have been considered irrelevant. For these reasons, the evolution of facultative-obligate symbiosis need to be treated separately from the evolution of specialist-generalist.

Theoretical studies that address the evolutionary transition toward obligate symbiosis are rare, but even there, this aspect is not the main focus. Frank [34] only focused on the evolution of cooperation, hence only mutualistic interactions were taken into account. Law and

Diekmann [7] considered parasitism but only evolution of vertical transmission was taken into account; horizontal transmission where symbionts have to experience the external environment was ignored. Van Baalen and Jansen [25], in an analysis of the mutualism-parasitism continuum, pointed out that members of associations are unlikely to give up their 'evolutionary sovereignty' as this requires their interests to be completely aligned, which they claim cannot occur whenever any of the partners has part of its lifecycle outside of the association. However, they did not consider trade-offs that linked traits expressed in the free and the associated states.

In this article, we attempt to fill in the gaps left by those studies. We will study the conditions under which evolutionary transitions to obligate symbiosis are favoured or disfavoured, assuming that there is a constraint on adaptations to the associations. We first unravel different fitness components for different lifestyles of a symbiont, then work out conditions whether the symbiont forgoes its independent reproduction or retains the capacity to reproduce when alone. A complete loss of independent reproduction is equivalent to obligate symbiosis, whereas nonzero independent reproduction suggests facultative symbioses with different levels of independent reproduction. Purely free-living organisms do not form associations with the hosts even though they are available.

To our knowledge, our study is the first to address the question of how facultative symbiotic relationships may become obligate. We therefore aim for a simple model that demonstrates the fundamental selective forces acting on this fundamental evolutionary transition. In order to do so, it is important to make some simplifications, for instance, we will leave out the host dynamics and consider the traits of the host as given. In this way, we can focus on the evolutionary dynamics of the symbiont and the types of constraints that might favour the evolution toward giving up independent reproduction. If even without the eco-evolutionary dynamics of the hosts, transitions toward obligate symbiosis are possible then host-symbiont interactions, such as host dynamics, types of ecological interactions, and so on, may not play a key role in such transitions. On the contrary, if transition to obligate symbiosis can only occur under limited conditions, then host coevolution may be essential. Furthermore, based on our simple model, we may obtain an idea of to what extent other factors may play a role in determining the evolutionary outcomes.

## The ecological dynamics

We focus on a population whose members can be either in a free-living state ($\mathcal{F}$) or associated with a host ($\mathcal{A}$) (Fig 1). It should be noted that we only consider long-term interactions such as lichen, insects and symbiotic bacteria, and so on, instead of the short-term interactions such as plant-pollinators or cleaning mutualisms. The free-living individuals produce free-living progeny at a rate $\rho$ while they die at a rate $\mu$. At a rate $\beta$, free-living individuals encounter hosts and form associations. Individuals in associations produce free-living progeny at a rate $\tau$, they die at a rate $\nu$. Bound individuals also reproduce together with their hosts at a rate $\sigma$; this is essentially the vertical transmission.

For the hosts we have to make a number of additional assumptions. We can assume, of course, that hosts have no appreciable dynamics of their own relative to the symbionts' dynamics, which sets vertical transmission automatically to zero. Alternatively we can impart various forms of population dynamics to the host. In the latter case, the symbionts may have various effects on host dynamics as well as induce vertical transmission. As in this study we aim to focus on the evolutionary pressures on the symbionts, we assume that the total host population size is a constant number $N$, and we will only consider host dynamics in a future study. Finally,

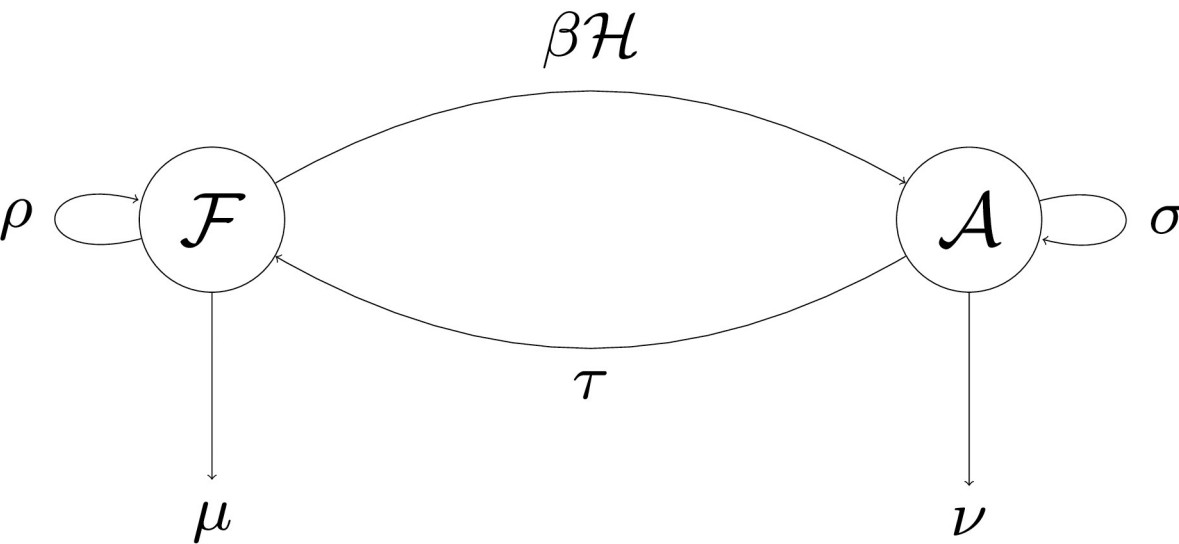

**Fig 1. Schematic representation of the model.**

we assume that if a host dies, its symbiont dies with it. In other words, dead hosts do not release free symbionts.

This gives us the dynamics of a mutant symbiont strain

$$\dot{\mathcal{F}} = \rho\mathcal{F} + \tau\mathcal{A} - \beta\hat{\mathcal{H}}\mathcal{F} - \mu(\hat{\mathcal{F}})\mathcal{F} \tag{1a}$$

$$\dot{\mathcal{A}} = \beta\hat{\mathcal{H}}\mathcal{F} + (\sigma - v)A \tag{1b}$$

where $\hat{\mathcal{F}}$ and $\hat{\mathcal{H}}$ stand for the equilibrium densities of the resident system (free-living residents, and free hosts, respectively) (S1 Appendix shows detailed calculation of feasible equilibrium). $\mu(\hat{\mathcal{F}}) = \mu_0(1 + c\hat{\mathcal{F}})$ indicates density dependent mortality with intensity $c$. We will also assume that $v > \sigma$ so that the population of associations will not grow without bounds because of high vertical transmission.

## A trade-off between reproduction and host encounter

In principle, mutation can affect any life history character, but we specifically consider mutations in the reproduction rates ($\rho$, $\tau$) and the host encounter rate ($\beta$) for two reasons: first, the most straightforward indication for obligate symbiosis is the forfeit of the reproduction in the free-living state; second, the encounter rate indicates the success rate of the formation of associations, and one can observe in nature diverse strategies of symbionts to encounter their hosts [35–38]. The values of the reproduction rate and the host encounter rate would therefore be good indicators of facultative and obligate symbiosis.

This set of assumptions effectively assumes that there is a trade-off between reproduction and host encounter. If $x$ represents an investment in reproduction, we can assume that

$$\rho'(x) > 0 \tag{2}$$

$$\tau'(x) > 0 \tag{3}$$

while

$$\beta'(x) < 0 \tag{4}$$

where the primes indicate the derivative with respect to $x$. The conditions above imply that if an individual spends its energy on increasing host encounter rate, its reproduction will reduce. Furthermore, being in association boosts bound reproduction so that

$$\rho(x) < \tau(x)$$

There could be other effects on survival and vertical transmission but for simplicity we will not consider these here, hence the only differences among strains is the three phenotypic parameters.

## Invasion analysis

A mutant, with dynamics (1), that adopts a different strategy than the resident is able to spread, replace the resident and become a new resident if its reproduction ratio $R_0$ is greater than one. We obtain the expression of $R_0$ using the next-generation method [39] (see S2 Appendix for detail). The invasion condition means that on average, a mutant individual is replaced by more than one mutant offspring, and it can be written as

$$\frac{\mathcal{C}_f(x) + \mathcal{C}_a(x)}{\mathcal{T}(x, x_r)} > 1 \tag{5}$$

where

$$\mathcal{T}(x, x_r) = \beta(x)\hat{\mathcal{H}} + \mu(\hat{\mathcal{F}}(x_r)) \tag{6}$$

is the overall decay rate of free-living individuals (so that $1/\mathcal{T}(x, x_r)$ represents the expected duration of the free-living state). The expected duration of the free-living state of the mutant depends on the population density of the resident $\hat{\mathcal{F}}(x_r)$, which itself depends on the strategy that the resident invest in its reproduction $x_r$.

There are two fitness contributions at the free-living state, independent reproduction,

$$\mathcal{C}_f(x) = \rho(x) \tag{7}$$

and through host encounter,

$$\mathcal{C}_a(x) = \beta(x)\frac{\tau(x)\hat{\mathcal{H}}}{v - \sigma} \tag{8}$$

Note that the contribution through host encounter depends on the availability of the unoccupied hosts, $\hat{\mathcal{H}}$, and the expected duration of an association $1/(v - \sigma)$. Condition (5) also suggests that without any trade-off, mutants that keep investing in both independent and bound reproduction $\rho(x)$, $\tau(x)$ will always be able to invade, suggesting that evolution of facultative symbionts that can reproduce independently in the absent of hosts is the only evolutionary end.

## Evolutionarily stable strategy

The loop where a new mutant emerges, spreads and replaces the resident then becomes a new resident will stop if there exists a resident population adopting a strategy $x^*$ such that mutants with other strategies cannot invade. This strategy is called the evolutionarily stable strategy (ESS) [40, 41]. When the resident population adopts the ESS, the selection gradient $dR_0(x, x_r)/$

$dx$ vanishes, which implies that the ESS $x^*$ satisfies

$$\frac{d\mathcal{C}_f(x)}{dx} + \frac{d\mathcal{C}_a(x)}{dx} = \left.\frac{\partial \mathcal{T}(x, x_r)}{\partial x}\right|_{x_r=x=x^*} \tag{9}$$

Our assumption that investing in host encounter rate comes at a cost on reproduction (condition 2, 3 and 4) implies that $d\mathcal{C}_f(x)/dx > 0$ and $\partial \mathcal{T}(x, x_r)/\partial x < 0$. Condition (9) is only satisfied if $d\mathcal{C}_a/dx < 0$. This means that mutants with higher investment in reproduction than the ESS resident undergo negative effects in their associated lifestyle, which strongly depends on how much cost is induced on their bound reproduction by investing on the host encounter rate (this is represented by $d\beta/d\tau = \beta'(x)/\tau'(x)$, where the primes indicate the derivatives with respect to the energy investment $x$).

To simplify the analysis, we assume that the symbiont makes the same investment in reproduction regardless of whether it is inside the host or in the external environment, i.e. $\tau'(x) = \rho'(x)$. This is not an unreasonable assumption, the difference between the bound reproduction and the independent reproduction may be due to how the symbiont exploits the host but not in how it invests in reproduction. But the mechanism underlying host exploitation is not within the scope of this model. Now, the existence of an ESS and its exact value depend on the magnitude of $\beta'(x)/\rho'(x)$ (Fig 2) [42, 43].

Condition (9) of a zero selection gradient can be converted into the condition for the intersection of the trade-off curve between host encounter rate and reproduction, and the invasion boundary (Fig 3), which leads to

$$\frac{\beta'(x)}{\rho'(x)} = \left.\frac{-\mathcal{M} - \hat{\mathcal{H}}\beta(x)\frac{\tau(\rho)}{d\rho}}{\hat{\mathcal{H}}(\tau(x) - \mathcal{M})}\right|_{x_r=x=x^*} \tag{10}$$

where $\mathcal{M} = \nu - \sigma$. The left-hand side represents the trade-off curve between independent reproduction and host encounter rate (Fig 2 and solid lines in Fig 3). The right-hand side represents the invasion boundary of a resident (dashed lines in Fig 3) which indicates the area in which mutants can invade the resident (we showed that this expression is in fact the slope of the invasion boundary in S3 Appendix). Residents that invest differently in reproduction may

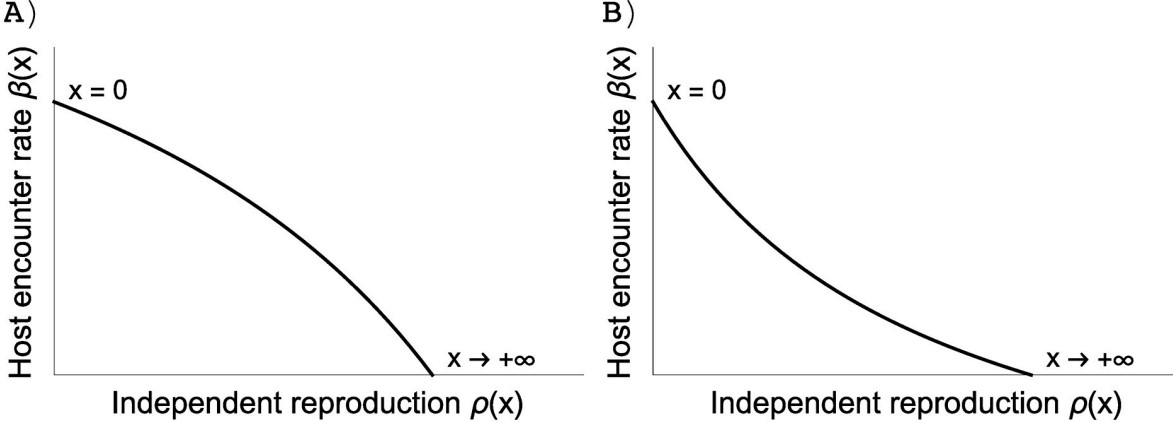

**Fig 2. Relationship between $\beta$ and $\rho$, which is derived from the assumption that increasing the investment $x$ in independent reproduction $\rho(x)$ leads to a reduction in host encounter rate $\beta(x)$.** Host encounter rate and independent reproduction are functions of the investment such that $\beta(x) = \beta_{max}/(\alpha x + 1)$; $\rho(x) = \omega x/(\delta x + 1)$. Depending on the parameters, the resulting trade-off can be convex (A) $\alpha = 0.7$, $\delta = 1.3$ or concave (B) $\alpha = 3.1$, $\delta = 1.3$. A convex trade-off curve indicates that investing in host encounter rate does not reduce much reproduction rate, whereas a concave trade-off curve indicates the reverse.

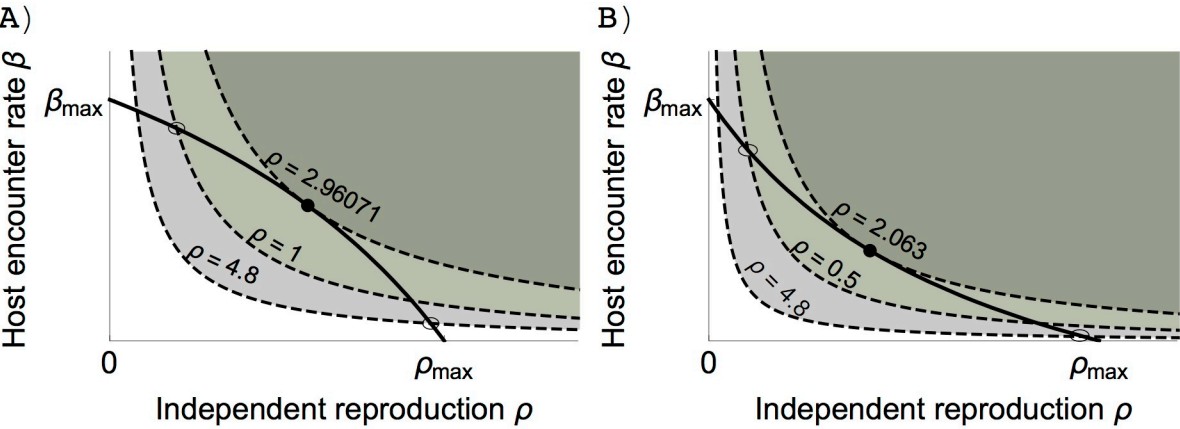

**Fig 3. Invasion boundaries as a function of different levels of resident investment in the independent reproduction rate are represented by dashed lines.** Open circles indicate the values of the traits of the residents. Shaded areas are invasion areas. Trade-off is assumed as in Fig 2. Free host population is $\hat{\mathcal{H}} = N - \hat{\mathcal{A}}$, where $N$ is the total number of hosts. ESS value is the tangent point (closed circle) between the invasion boundary and the convex trade-off curve (A) and concave trade-off curve (B). Parameter values: $b = 3$, $v = 3$, $c = 2$, $\sigma = 0$, $\mu_0 = 3$, $\beta_{max} = 5$, $\omega = 6.5$, $N = 100$, (A) $\alpha = 0.7$, $\delta = 1.3$, (B) $\alpha = 3.1$, $\delta = 1.3$.

have different invasion boundaries, and condition (10) suggests that the ESS has to be the tangent point of the trade-off curve and the invasion boundary (closed circles in Fig 3) because at this point, at least locally, the invasion area contains no possible mutant. Rewriting the selection gradient in this way is convenient because we can disentangle the effect of the trade-off from the ecological feedback on the value of the ESS.

Depending on the conditions, the ESS can be obligate symbiosis, facultative symbiosis, or completely free-living organisms. A larger value of $\beta'(x)/\rho'(x)$ may lead to an ESS of smaller independent reproduction. In other words, if investing in host encounter comes at a high cost, symbionts may evolve to forgo their independent reproduction, because the amount of energy left to invest in its reproduction is too small. However, a high cost of investment in host encounter does not always mean an increase in host dependency because the ESS value also depends on the ecological feedback, such as the abundance of free hosts.

Given a fixed cost of investment in the host encounter rate, i.e. fixed value of $\beta'(x)/\rho'(x)$, if all else being equal, strategies that result in a high bound reproduction value (i.e. high value of $\tau(x)$) will lead to higher dependency of the symbiont on the host. There are several ways to obtain a high bound reproduction value, one of which is exploiting resources from the hosts (Fig 4). Thus, starting with a population of facultative symbiont, the higher the benefits inside the association obtained by exploiting the hosts, the more the symbiont is willing to forgo its independent reproduction and becomes more dependent on the host (Figs 4 and 5).

However, an ESS of fully obligate symbiosis is not inevitable, if not often difficult to reach. It is only possible if the zero independent reproduction is the ESS, or the strategy $x^*$ such that $\rho(x^*) = 0$ satisfies

$$\begin{cases} \left(\dfrac{\beta'(x)}{\rho'(x)} < \dfrac{\mathcal{M} + \hat{\mathcal{H}}\beta(x)\frac{d\tau}{d\rho}}{\hat{\mathcal{H}}(\mathcal{M} - \tau(x))}\right)\Bigg|_{x=x^*} \\ \tau(x) > \mathcal{M}|_{x=x^*} \end{cases} \tag{11}$$

Condition (11) suggests that obligate symbiosis is an ESS only if the bound reproduction $\tau(x)$ is sufficiently large, which indicates that the benefits that the symbionts gain from the host must be high enough (Red area in Fig 5). If the resident of obligate symbionts do not

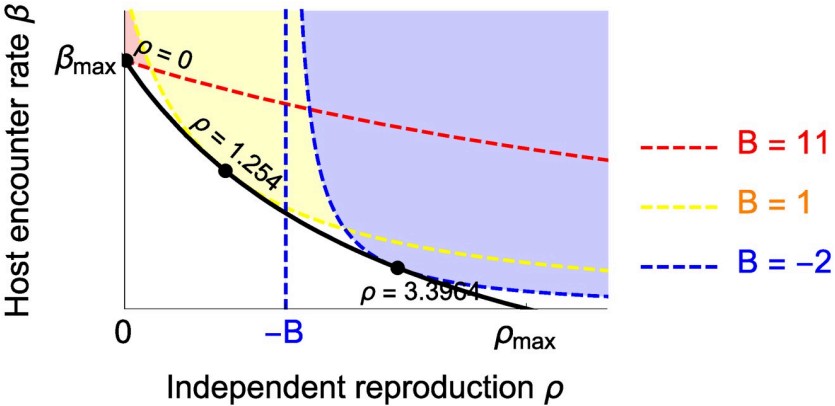

**Fig 4. Effect of the compound benefit ($B = b − v + \sigma$) on the ESSs in the case of a concave trade-off curve (black line) under the assumption that the bound reproduction is $\tau(x) = \rho(x) + b$, where $b$ is the bonus reproduction by exploiting the host.** Trade-off and host population are as in Fig 3. Dashed lines are the invasion boundaries that separate shaded areas in which mutants can invade and blank areas in which mutants cannot. Black points indicate the ESS values that are corresponded with different values of $B$. Other parameters: $c = 2$, $N = 100$, $\mu_0 = 3$, $\omega = 6.5$, $\beta_{max} = 5$, $\alpha = 3.1$, $\delta = 1.3$, $\sigma = 0$.

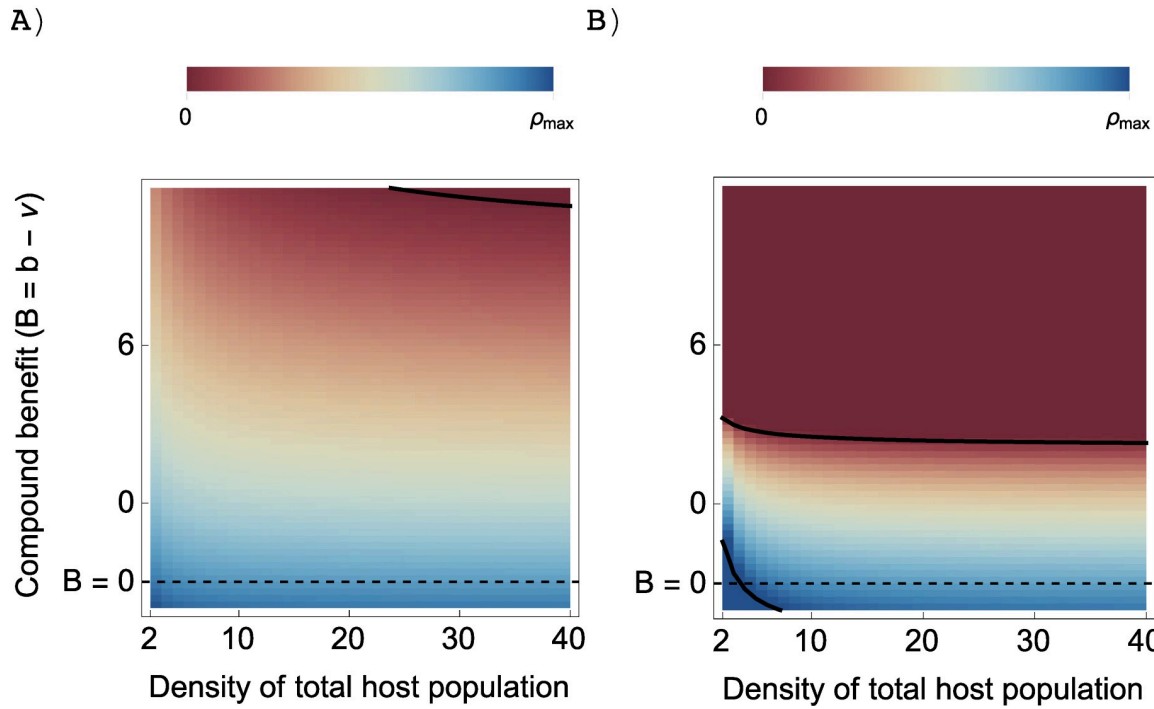

**Fig 5. Effect of the compound benefit $B$ and density of the total host population $N$ on the transition from fully retaining independent reproduction ($\rho = \rho_{max}$) to completely forgoing independent reproduction ($\rho = 0$) under cases with no vertical transmission ($\sigma = 0$).** Trade-off is as in Fig 3. A) Convex trade-off curve where $\alpha = 0.7$, $\delta = 1.3$, B) concave trade-off curve where $\alpha = 3.1$, $\delta = 1.3$. The dashed line corresponds to the case when the symbionts gain no benefit ($b = 0$). The thick lines separate the three regions: pure free-living (dark blue), obligate symbiosis (dark red) and faculative symbiosis with varying degrees of independent reproduction.

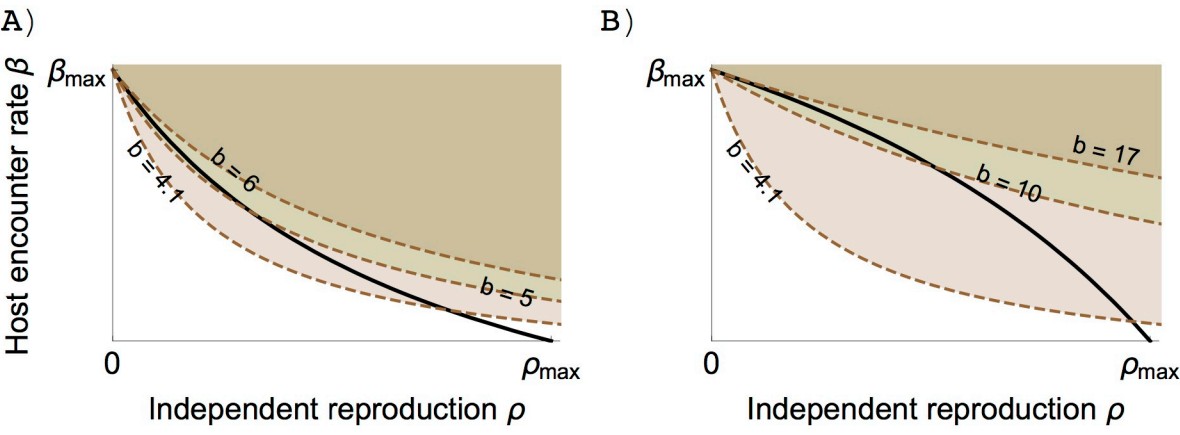

**Fig 6. Different invasion boundaries (dashed lines) of the residents with zero independent reproduction and different values of the reproduction bonus $b$, assuming that $\tau(x) = \rho(x) + b$.** Trade-off and other conditions are similar to Fig 3. The higher the bonus reproduction, the less negative the slope of the invasion boundary compared with the slope of the trade-off curve (black line). A) Concave trade-off $\alpha = 3.1$, $\delta = 1.3$. B) convex trade-off $\alpha = 0.7$, $\delta = 1.3$. Other parameter values $N = 10$, $c = 2$, $\sigma = 0$, $v = 3$, $\mu_0 = 3$, $\beta_{max} = 5$, $\omega = 6.5$.

reproduce sufficiently via the hosts, mutants that invest more in the independent reproduction will be able to invade (Fig 6).

At the other end, returning to fully free-living organisms is no less easier, which will only happen when

$$\tau(x) \leq \mathcal{M}|_{x_r=x=x^*=0} \tag{12}$$

$$\tau(x) > \mathcal{M}|_{x_r=x=x^*} \text{ and } 0 > \frac{\beta'(x)}{\rho'(x)} > \frac{\mathcal{M}}{\hat{\mathcal{H}}(\mathcal{M} - \tau(x))}\bigg|_{x_r=x=x^*} \tag{13}$$

with $\beta(x^*) = 0$. Condition (12) suggests that if reproducing via the association is small, especially much smaller than the mortality in the associations (i.e. $\tau(x) \ll v$) then there is no point in investing in this lifestyle, and becoming free-living can be an evolutionarily stable point (Dark blue area below the dashed line in Fig 5B). Returning to the fully free-living lifestyle may be possible even if the symbionts see benefits in the associated lifestyle, i.e. sufficiently large $\tau(x)$, but only when hosts are extremely scarce, because only when $\hat{\mathcal{H}} \approx 0$ can condition (13) be satisfied (Fig 7 and dark blue area above the dashed line in Fig 5B).

Our analysis suggests that intermediate strategies of independent reproduction are common evolutionary ends, suggesting that facultative symbiosis should be common. The only way for full symbiosis requires a high reproduction value inside the association, which could be obtains by high exploitation from the hosts, and this in itself very likely depends on host co-evolution. How exactly does the symbionts exploit the hosts deserves a separate and more thorough study, and will not be dealt with here. Even so, we will briefly discuss possibilities how a symbiont could increase its bound reproduction.

## Uncoupling free and bound reproduction on the difficult evolution toward obligate symbiosis

One of the important condition for an ESS of obligate symbiosis is that the symbiont has to gain sufficient bound reproduction $\tau(x)$. There are many way in which a symbiont can increase this value, and one of the simplifying assumptions we made for the numerical analysis example is that the rate of reproduction of associated symbionts is the same as that of free-living individuals plus a bonus they receive from their partner $\tau(x) = \rho(x) + b$. Our assertion that obligate

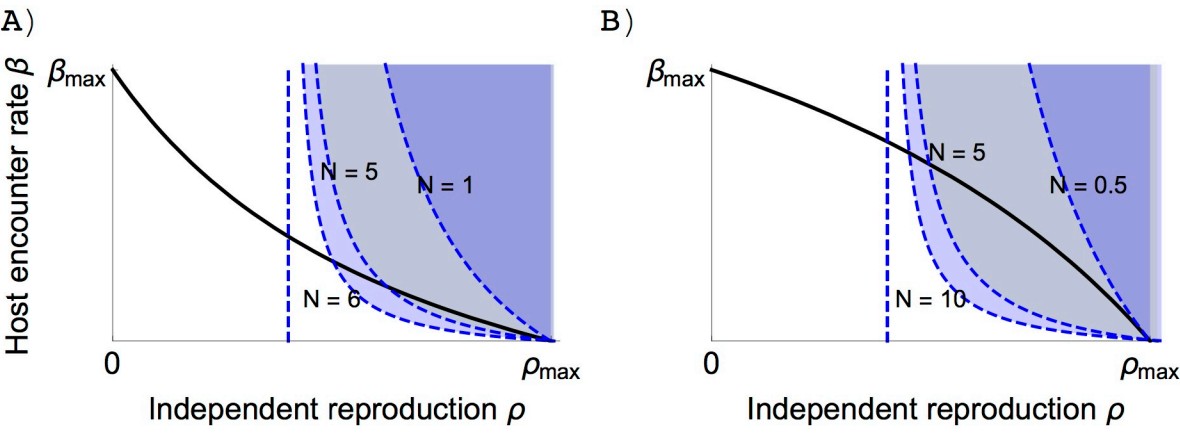

**Fig 7. Different invasion boundaries (dashed lines) of the residents with zero host encounter rate in the environment of different total number of hosts N.** The lower the density of the total host population, the more negative the slope of the invasion boundary (dashed lines) compared with the slope of the trade-off curve (black line). A) Concave trade-off $\alpha = 3.1$, $\delta = 1.3$. B) convex trade-off $\alpha = 0.7$, $\delta = 1.3$. The blue lines are the invasion boundaries that respectively corresponds to different values of density of total host population $N$. Other parameter values $b = 1$, $c = 2$, $\sigma = 0$, $v = 3$, $\mu_0 = 3$, $\beta_{max} = 5$, $\omega = 6.5$.

symbionts give up free reproduction thus implies that the rate of reproduction of symbionts is fully determined by the host. This could of course be the actuality in many full-blown obligate symbioses. For instance, the replication rate of mitochondria is tightly controlled by their host's nucleus [44]. However, this cannot be the full picture. Many parasites, that are full-blown obligate symbionts, strongly resist the efforts of their hosts to curb their reproduction. Therefore, our assumption of identical reproduction rates and a fixed bonus is not justified in many cases.

The first possibility is that symbionts reproduce differently in different environments (so that $\tau(x) \neq \rho(x) + b$ for a given $x$). This could be either due to differential resource densities or that the individuals are able to detect that they are in different environments and respond to it. Evidence that symbionts behave differently inside and outside of hosts has been documented [31, 45]. A consequence of this relaxation is that $\rho'(x) = \tau'(x)$ no longer holds, therefore $d\tau/d\rho$ is no longer a constant, which greatly simplified our analysis.

A more in-depth analysis of the consequences of decoupling the reproduction rates $\rho(x)$ and $\tau(x)$ will be published elsewhere, but suffice it here to state that it does not affect our conclusion that constrains performances in the free state and in the associated state are central to our understanding of what governs the evolutionary transition to obligate symbiosis.

A second, equally interesting possibility is that the benefits are somehow dependent on the symbionts' investment in reproduction. This is reflected in numerous cases where mutualistic symbionts trade some of their resources with their partners. Then, $b$, the bonus reproduction gained from the exploiting the host, is no longer a constant but depend on $x$. Moreover, this involves consideration of how the host responds to the symbiont, and thus of coevolution. This falls outside of the scope of this article. However, it is easy to see that if $b$ depends positively on $x$ it will most likely raise the invasion boundary and thus select for increased values of the encounter rate $\beta$ generally.

## Discussion

The term symbiosis was introduced in the late 1870s after several studies on lichens had shown that an individual lichen consists of individuals of more than one species [6]. Symbiosis was then considered as a source of evolutionary innovation ([6] and [46] and their references).

Symbiosis of prokaryotic cells is suggested to result in eukaryotic cells [3], which is considered one of the major evolutionary transition by Maynard Smith and Szathmary [47], and this is only one extreme case of obligate symbiosis in which obligate dependency is required from both partners. Examples of obligate and facultative symbioses are ubiquitous, with different levels of dependency from both partners. Yet, how obligate symbioses may evolve out of facultative ones is not well-understood, thus studying this intriguing transition process may give insights and guide for future studies.

Our study shows that it is useful to consider facultative and obligate symbiosis not as discrete traits but as part of a continuum where the key trait is the capacity of independent reproduction, that is, the capacity of the symbiont to reproduce without the host. The continuum of the independent reproduction implies that there are varying degrees of dependency in facultative symbiosis (this has been realised also in [48]), and only symbionts that completely give up their independent reproduction should be considered obligate symbionts.

Our model suggests that, if encountering hosts comes at a price of reproduction, gaining benefits from the symbiotic lifestyle will select for more investment in host searching and loss of independent reproduction. Moreover, the higher the price, the less likely facultative symbiosis will be an evolutionary outcome, much the same as generalists are not necessarily favoured when the generalist-specialist trade-off curve is concave [49–52]. However, our model shows that even with a high cost of reproduction, a complete loss of independent reproduction is not guaranteed unless benefits from being in association are sufficient. This is because we assume that a symbiont invests in its free reproduction the same way that it does in its bound reproduction; what eventually keeps the symbiont in the association is the reproduction bonus that it receives from its host, which is assumed to be a constant amount in the scope of our model.

There may be two reasons that the symbionts invest the same in reproduction regardless of whether it is free-living or associated with its host. First, the symbiont is not sensitive enough to perceive the differences between the external and the host environment. Second, the two environments are so similar that the symbiont does not need to respond differently. For either reason, obligate symbiosis is unlikely to evolve. In contrast, if the symbiont behaves differently in the host environment than in the external environment, regardless of whether this difference is a result of active response from the symbiont or it is simply due to strong intrinsic dissimilarities of the two environments, then obligate symbiosis may be a possible outcome.

Ecological feedback on the density of free hosts may also make the full transition toward obligate symbiosis difficult. As we did not take into account host dynamics, higher benefits also imply higher density of associations, which leads to a lower density of free hosts. Therefore if an increase of benefits from the association cannot compensate for the reduction in the number of free hosts, obligate symbiosis will not be an evolutionarily outcome. Even if we relax the assumption of host dynamics, ecological feedback can take place in the competition between associations and free hosts, and similar results can be obtained.

The evolution of a pure free-living lifestyle may not be easy either as passive formation of associations may be difficult to avoid. While we do not have evidence for whether or not it is easy to gain some benefits in the associations, it is certainly difficult to have no interaction at all with other species that may serve as potential hosts. Our model thus suggests that facultative symbiosis should be common but how then do we explain the many examples of obligate symbioses?

First, many symbioses may be less obligate than they seem. In fact, partners in many symbioses that appear to be obligate in nature can nevertheless often been shown to be able to grow and reproduce outside the associations in laboratories [53–55]. Second, in many cases partners in profitable symbioses do need to make a significant effort to meet a suitable host. Our model also suggests that symbionts that have to make a bigger effort to find hosts are more likely to

sacrifice all their independent reproduction while those that depend more on random encounter are more likely to retain some independent reproduction. There are examples indicating that such a pattern occurs in nature.

Bacterial pathogens such as *Bacillus cereus* and *Bacillus anthracis* both form endospores as a protection strategy against destruction of gastric acid and harsh environmental conditions. This strategy is suggested to facilitate the transmission of the parasites. *B. anthracis* is a well-known obligate pathogen that cannot reproduce outside hosts [56] while *B. cereus* is present in the external environment under both spore and vegetative forms but proliferation seems rare [57, 58] (only one experimental study showed that the bacteria can germinate and grow in a soil mimicking condition [59]). According to our model, they can be placed at the obligate symbiotic end.

On the other hand, many other bacterial pathogens such as *Salmonella spp.*, *Listeria monocytogenes*, and *Vibrio cholerae* enter a stationary phase and depend on dose to overcome the gastric acid [30, 60, 61]. It is not unlikely that this strategy is less costly than forming spores because spores are often assumed to resist more extreme conditions for a longer period of time than a stationary phase does. Interestingly, *Salmonella*, *L. monocytogenes*, and *V. cholerae* have been shown to retain the ability to reproduce in the external environment [30, 62, 63], or equivalently, they are more likely at the freeliving-facultative symbiotic end.

Another class of the effect of transmission costs is provided by mutualistic symbioses. Rhizobia and Arbuscular Mycorrhizas (AM) are symbionts that form nodules in plant roots; the former are bacteria that do not form spores while the latter are fungi that reproduce via sporulation. Both Rhizobia and AM encounter hosts randomly and establish the symbiotic relationship via complex molecular signals [31]. Intuitively, AM may increase the chance to encounter with their hosts by forming spores that can withstand harsh environmental conditions, whereas Rhizobia may die when the host is scarce and the environment is unfavourable. When environmental conditions are appropriate, Rhizobia are found as free-living bacteria in soil while AM may pay too high a price for sporulation that they cannot reproduce without the host plant even in laboratories.

However, transmission costs may not explain everything. For instance, the parasitic bacterium *Campylobacter jejuni* survives through gastric acid without forming spores and its successful transmission is dose dependent, thus it is expected that *C. jejuni* is capable of independent reproduction. Yet, *C. jejuni* is extremely fragile and seems unable to reproduce in the external environment [64]. In a similar fashion, bacteria of the genus *Shigella* seem able to survive but not to reproduce in the external environment [65]. They do not form spores yet are more efficient than *E. coli* or *Samonella* in withstanding gastric acid owing to their sophisticated mechanisms of acidic resistance [66].

That *C. jejuni* and *Shigella* cannot reproduce without hosts may simply imply that the benefit from the association is sufficient for the organisms to give up all the independent reproduction, or that spore forming is not the only costly aspect of host encounter. For instance, Ectomycorrhizas (ECM), fungi that form symbioses with plant roots, do form spores; they are not found to reproduce without host in the environment but can be cultured *in vitro*. The main difference between AM and ECM is that AM penetrate the root cells whereas EM do not [67]. If all else is equal, that is, if the cost to produce molecular signals and penetrating the epidermis is similar between the two types of mycorrhizas then it is possible that the intracellular colonization of AM is more costly than the extracellular establishment of ECM.

These examples show that it is necessary to better quantify the costs of host encounter and independent reproduction if we want to understand the conditions that favour obligate symbiosis. The relationship between host encounter and independent reproduction may provide guidance for studies of symbionts whose lifecycles are not well understood.

One of the limitations in our model is that it overlooks the ecological interactions between hosts and symbionts, as it assumes that the total host density is fixed. These interactions are nevertheless likely to have strong effects on the evolution of both the hosts and the symbionts [68–70], and are expected to affect the level of independent reproduction. Van Baalen and Jansen [25] showed that mutualistic associations may select for partners to forgo independent reproduction because mutualism increases the alignment of interest. Future work adding the effect of ecological interactions will shed more light on the evolutionary transitions of the free-living-obligate continuum. For instance, are obligate mutualistic symbioses more likely to evolve than obligate parasites or not?

We proposed a simple model, thus the mathematical results are straightforward and intuitive, yet interesting biological interpretations can be inferred from it. More importantly, it will give useful insights into possible evolutionary outcomes should more complicated models be built upon. In a future study, we will study the effects of additional mechanisms which are know to be important, such as incorporating host dynamics, considering more general trade-off, taking into account ecological interactions and so on.

## Supporting information

**S1 Fig. Bifurcation analysis of the ecological equilibrium.**
(TIF)

**S1 Appendix. Stability of the resident population equilibrium.**
(PDF)

**S2 Appendix. Next generation method.**
(PDF)

**S3 Appendix. Conditions for intermediate evolutionarily stable strategy (ESS).**
(PDF)

**S4 Appendix. Effect of the reproduction bonus b on the associated population at equilibrium.**
(PDF)

**S1 Data.**
(NB)

**S2 Data.**
(NB)

## Author Contributions

**Conceptualization:** Phuong Linh Nguyen.

**Formal analysis:** Phuong Linh Nguyen, Minus van Baalen.

**Investigation:** Phuong Linh Nguyen.

**Software:** Phuong Linh Nguyen.

**Supervision:** Minus van Baalen.

**Writing – original draft:** Phuong Linh Nguyen.

**Writing – review & editing:** Minus van Baalen.

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
