## [Decision Letter · Decision Letter 0]

7 Nov 2019

PONE-D-19-25560

On the difficult evolutionary transition from free-living lifestyle to obligate symbiosis

PLOS ONE

Dear Mrs Nguyen,

Thank you for submitting your manuscript to PLOS ONE. After careful consideration, we feel that it has merit but does not fully meet PLOS ONE’s publication criteria as it currently stands. Therefore, we invite you to submit a revised version of the manuscript that addresses the points raised during the review process.

The reviewer, who is expert in evolutionary conflicts of interest, found difficulty following many of the arguments, and their relationship to the biology. Therefore please attempt to clarify these in response to the reviewers comments, as far as possible.

We would appreciate receiving your revised manuscript by Dec 22 2019 11:59PM. To enhance the reproducibility of your results, we recommend that if applicable you deposit your laboratory protocols in protocols.io, where a protocol can be assigned its own identifier (DOI) such that it can be cited independently in the future. For instructions see: http://journals.plos.org/plosone/s/submission-guidelines#loc-laboratory-protocols

We look forward to receiving your revised manuscript.

Kind regards,

James A.R. Marshall, BSc, PhD

Academic Editor

PLOS ONE

Journal Requirements:

2. Thank you for providing your Data availability statement 'The data underlying the results presented in the study are obtained from model simulation. The codes that generated the data are available upon demanding the author. Email: lpnguyen@biologie.ens.fr'. In this instance, in order to ensure reproducibility, we would encourage you to share either the data or the codes with the manuscript or in a repository. Please also note that PLOS ONE typically expects data to be held by a named data access committee or named ethics committee or similar, and that it is not acceptable for an author to be the sole named individual responsible for ensuring data access (https://journals.plos.org/plosone/s/data-availability).

"No".

i) Please complete your Competing Interests on the online submission form to state any Competing Interests. If you have no competing interests, please state "The authors have declared that no competing interests exist.", as detailed online in our guide for authors at http://journals.plos.org/plosone/s/submit-now

ii)  This information should be included in your cover letter; we will change the online submission form on your behalf.

"This work has received support under the program « Investissements d’Avenir » 492

launched by the French Government and implemented by ANR with the references 493

ANR-10-LABX-54 MEMOLIFE and ANR-11-IDEX-0001-02 PSL* Research University".

"Linh Phuong NGUYEN

Contrat Doctoral no2016-2

Ecole Normale Superieur

PSL Research University

https://www.ens.fr/en

https://www.psl.eu/en

The sponsors do not play any role in the study design, data collection and analysis, decision to publish or preparation of the manuscript".

Reviewers' comments:

Reviewer's Responses to Questions

**Comments to the Author**

1. Is the manuscript technically sound, and do the data support the conclusions?

Reviewer #1: Partly

2. Has the statistical analysis been performed appropriately and rigorously? 

Reviewer #1: Yes

3. Have the authors made all data underlying the findings in their manuscript fully available?

Reviewer #1: Yes

4. Is the manuscript presented in an intelligible fashion and written in standard English?

Reviewer #1: No

5. Review Comments to the Author

Reviewer #1: In this paper, the authors analyze and describe a model of the evolution of the transition from free-living to obligate symbiosis. Unfortunately I found the paper hard to follow and at a certain point I just got lost. I would be happy to review the article again if it can be substantially clarified.

Abstract: I’m confused by the first two sentences. First sentence: Who has suggested this? Endosymbiosis of e.g. mitochondria is often hypothesized to have resulted from a predator-prey scenario, in which case it would not have evolved from what we normally think of as facultative symbiosis. Second sentence: Would e.g. protomitochondria, once engulfed, have the choice of living independently? I feel as though there is one obvious way in which individuals would “give up” the ability to live freely: the organisms which enter into certain symbioses outcompete those which do not, there are payoffs for closer and closer association (e.g. removal of conflict, better exchange of resources), and over time natural selection acts to remove those stretches of DNA which encode for free living, which are no longer needed and hence are a selective burden. So is this really a riddle? (line 15)

line 2: “contact”?

27–98: This may be a matter of taste, but I feel that a lot of this material going over existing models probably belongs in the discussion. I found it hard to work out from the introduction what the main message of the paper is.

39: This paper is just by Frank, not Frank and colleagues.

48: “sacrifices”

81: “complete”

100–104: There are lots of different kinds of symbiosis in the world and I think it would help to bring in some concrete examples here to clarify what you are modelling. Does this model apply only to endo/ectosymbiosis? Does it also apply to mutualisms like plant-pollinator and cleaner symbioses? I think it’s more about ecto/endosymbiosis, focusing on the symbiont rather than the host, but because the introduction was quite sweeping—e.g. citing in lines 4–9 previous work that looks very broadly at e.g. mutualisms like cleaner-client and plant-pollinator—it’s hard to grasp what’s being modelled here. Also, the parameter sigma needs to be explained here: when an A makes another A at rate sigma, is the new A associated with the same host as its parent or a new one?

105–112: So this model cannot analyze vertical transmission? If we are indeed focusing on endo/ectosymbioses, isn’t vertical transmission a rather important part of the evolution of obligate symbioses?

117–118: Does this assumption reduce the generality of your model? Why was this assumption used to prevent the population of associations from growing without bound, instead of e.g. a saturating growth function?

136–205: These sections need to be substantially clarified. Moreover, I feel that statements like 167-169 and 190-192, which restate results as statements about geometry, are not helpful. I would appreciate these results being explained in more biological terms. I mostly gave up on the results section at this point.

361–364: Unclear what this sentence means.

409–411: The fact that Trichinella can be induced to grow on intestinal epithelial cells in a lab has no bearing on how “obligate” its symbiosis is in nature. Particularly as the intestine forms part of its natural habitat. What matters is what happens in nature.

414: “sacrifice”

Figures, generally: Please provide labels A and B on all the plots, and please left-align, don’t right-align these labels. I have never seen this done before and it’s very confusing. All the plots should have informative axis titles like Fig. 5 so they can more easily stand alone.

Fig. 2: From the caption and figure alone, it’s unclear whether this trade-off is a result or an assumption. Obviously this is stated in the text, but it just doesn't help comprehension to have this much ambiguity.

I’m sorry I couldn’t be more helpful on this occasion.

Best wishes,

Nick Davies

London School of Hygiene and Tropical Medicine

6. PLOS authors have the option to publish the peer review history of their article (what does this mean?). If published, this will include your full peer review and any attached files.

Reviewer #1: No

---

## [Author Response · Author response to Decision Letter 0]

30 Dec 2019

Dear Nick Davies,

Thank you for reviewing our manuscript. You will find below our response to each point that you raised. We hope that they sufficiently clarify your questions.

# Abstract: Im confused by the first two sentences. First sentence: Who has suggested this? Endosymbiosis of e.g. mitochondria is often hypothesized to have resulted from a predator-prey scenario, in which case it would not have evolved from what we normally think of as facultative symbiosis. Second sentence: Would e.g. protomitochondria, once engulfed, have the choice of living independently? I feel as though there is one obvious way in which individuals would give up the ability to live freely: the organisms which enter into certain symbioses outcompete those which do not, there are payoffs for closer and closer association (e.g. removal of conflict, better exchange of resources), and over time natural selection acts to remove those stretches of DNA which encode for free living, which are no longer needed and hence are a selective burden. So is this really a riddle? (line 15)

Response: The idea that obligate symbiosis evolves from facultative symbiosis is our extension of Poulin’s idea in his book: ”Evolutionary Ecology of Parasites” [1]. Here, he suggested that obligate parasites may evolve from facultative parasites if the contact between the parasite and the host is frequent enough [1, Chapter 2]. We argue that the type of ecological interactions may not play a key role in this evolutionary transition, that is, obligate symbiosis may evolve from facultative symbiosis, regardless of whether the interaction is parasitism or mutualism.

We agree that for many symbiotic systems, the associations may have resulted from prey-predator interactions, and if the engulfed symbionts are then maintained purely by vertical transmission with the host then it is obvious that obligate symbiosis can evolve directly from free-living organisms. However, the prey-predator interaction at the level of microorganisms may completely be different from the interaction at the level of macroorganisms. For instance, a deer is surely dead and cannot escape after being consumed by a tiger, but a bacteria that are engulfed inside a host may escape. In fact, some hydra strains form association with algae by engulfing the algae, but the algae symbionts have been shown to be able to escape from the hydra hosts and enter the external environment ([2]). We argue that the mode of acquiring the symbionts is not that important, and symbiosis formed via engulfing is no different from symbiosis formed via other modes such as rhizobia entering the plant through forming nodules in the roots, or Vibrio fisherie entering the bobtail squid in the specialised light organ. Therefore, we find that conventional facultative symbiosis still holds even when it is formed via prey-predator interaction.

In the example that you raised, when protomitochondria were engulfed, it does not follow by being trapped inside their bacterial hosts forever and could only vertically transmit with the hosts and benefit from the hosts. If for any reason, the protomitochondria can escape from the host, they have to deal with the external environment, such as harsh environment, intra and interspecific competition, then they should be better off retaining the ability of independent reproduction, thus, evolving toward obligate symbiosis may not seem so obvious.

Below is our modified paragraph. It can be found in line 22-32 in the revision text file.

” Obligate symbionts have to evolve from free-living individuals, regard- less of whether they are parasites or mutualists, because if there were no free-living individuals to begin with, there would be no ingredients for associations. If this process occurs due to, for instance, an event of engulfment of the symbiont by the host, followed by purely vertical transmissions of the symbionts to maintain the population then obligate symbionts may evolve directly from free-living organisms. However, many symbionts have to experience external environment before encountering new hosts, and facultative symbiosis is much likely an intermediate step leading toward obligate symbiosis. ”

# line 2: contact?

Response: This has been corrected in line 2

#27-98: This may be a matter of taste, but I feel that a lot of this material going over existing models probably belongs in the discussion. I found it hard to work out from the introduction what the main message of the paper is.

Response: We slightly modified the structure of the introduction, and shorten the explanation of existing models. Below is our modified paragraph. It can be found in line 64-74.

”Theoretical studies that address the evolutionary transition toward ob- ligate symbiosis are rare, but even there, this aspect is not the main focus. Frank [34] only focused on the evolution of cooperation, hence only mutualistic interactions were taken into account. Law & Dieckmann [7] considered parasitism but only evolution of vertical transmission was taken into account; horizontal transmission where symbionts have to experience the external environment was ignored. Van Baalen & Jansen [25], in an analysis of the mutualism-parasitism continuum, pointed out that members of associations are unlikely to give up their evolutionary sovereignty’ as this requires their interests to be completely aligned, which they claim cannot occur whenever any of the partners has part of its lifecycle outside of the association. How- ever, van Baalen & Jansen [25] did not consider trade-offs that linked traits expressed in the free and the associated states.”

# 39: This paper is just by Frank, not Frank and colleagues. 

Response: This has been corrected in line 65.

# 48: sacrifices

Response: This part has been removed

# 81: complete

Response: This has been corrected in line 80

# 100 - 104: There are lots of different kinds of symbiosis in the world and I think it would help to bring in some concrete examples here to clarify what you are modelling. Does this model apply only to endo/ectosymbiosis? Does it also apply to mutualisms like plant-pollinator and cleaner symbioses? I think its more about ecto/endosymbiosis, focusing on the symbiont rather than the host, but because the introduction was quite sweeping e.g. citing in lines 49 previous work that looks very broadly at e.g. mutualisms like cleaner- client and plant-pollinatorits hard to grasp whats being modelled here. Also, the parameter sigma needs to be explained here: when an A makes another A at rate sigma, is the new A associated with the same host as its parent or a new one?

Response: First, we affirm that our model only applies to endo/ectosymbiosis, such as algae-fungi in lichen, aphid-Buchnera, Rhizobia-legume, and so on, but not to the relationships like plant-pollinator.

Furthermore, we completely agree with the reviewer that the definition

of symbiosis is rather ambiguous. When symbioses were discovered in the late 19th centuries, de Bary defined symbiosis as ”the living together of unlike named organisms”. According to this definition, symbiosis includes both parasitism, mutualism, and associations of all level of partner dependency, from endo/ectosymbiosis, such as lichen, and insects-bacteria to short term symbiosis such as cleaner-client and plant-pollinator. However, it is not very clear whether de Bary did want to include associations such as plant- pollinator or cleaning symbiosis. Zook [3] did mention this ambiguity and redefined symbiosis, that is ”Symbiosis is the acquisition of an organism(s) by another unlike organism(s), and through subsequent long-term integration, new structures and metabolism(s) emerge”. The symbiotic system that we modelled is closer system defined by Zook [3], that is, it includes only endo/ectosymbiosis.

Furthermore, the parameter σ, vertical transmission is only relevant in the context of endo/ectorsymbiosis. When an association (A) makes another A, the symbiont reproduces with the host, that is the new A is the association of a new symbiont and new host.

It is important to note that in terms of the type of interactions, we still follow de Bary’s definition, that is we consider both parasitism and mutual- ism, even though the type of interactions cannot be taken to account in this model because we considered a fixed number of hosts for simplification.

We also add our affirmation when explaining the model schematic, which can be found in line 100-102.

” It should be noted that we only consider long-term interactions such as lichen, insects and symbiotic bacteria, and so on, instead of the short-term interactions such as plant-pollinators or cleaning mutualisms. ”

# 105 - 112: So this model cannot analyze vertical transmission? If we are indeed focusing on endo/ectosymbioses, isnt vertical transmission a rather important part of the evolution of obligate symbioses?

Response: We agree with the reviewer that vertical transmission may play an important part in the evolutionary transition toward obligate symbioses and it would be best if we could consider both types of transmissions. However, for various reasons, we focus on horizontal transmission instead of vertical transmission. First, in nature, there are many examples of obligate symbioses with mixed modes of transmission [4]; in fact, many obligate parasites are only horizontally transmitted (it should be noted that we only consider symbionts with simple life cycle, that is symbionts with only one host). Secondly, early in the evolutionary transition from facultative to ob- ligate symbiosis, vertical transmission is likely to be rare as it may require specific adaptations [5]. Therefore, considering the evolution of both types of transmission will make our model much more complicated while we are aiming for a simple model as a start to understand this transition.

# 117 - 118: Does this assumption reduce the generality of your model? Why was this assumption used to prevent the population of associations from growing without bound, instead of e.g. a saturating growth function?

Response: We do not think that the assumption that the vertical transmission is smaller than the bound mortality rate (σ < ν) will reduce the generality of our model because as we responded in the question above, vertical transmission is very likely rare and only happens by chance in the early evolutionary transition. Examples of exclusive vertical transmission that are observed in nature are mostly systems that are already obligate symbioses, and we cannot be sure whether vertical transmission leads to obligate symbiosis or is it because obligate symbiosis (with horizontal transmission) promotes the evolution of vertical transmission.

The assumption that σ < ν is to prevent the population of A from growing exponentially when the host encounter rate β evolves toward 0. This is the case of completely free-living population because even when there are free-living symbionts and hosts around, the symbionts will not encounter and establish inside the host and form association. If the mutant of zero β is created through horizontal transmission then nothing will happen but if it is created via vertical transmission, then the population of association A will grow exponentially. A saturating growth function require at least quadratic function with regard to the association population, and will prevent analytical results of the equilibrium.

# 136 - 205: These sections need to be substantially clarified. Moreover, I feel that statements like 167-169 and 190-192, which restate results as statements about geometry, are not helpful. I would appreciate these results being explained in more biological terms. I mostly gave up on the results section at this point.

Response: The entire section of model analysis and result have been modified in line 140-234.

# 361 - 364: Unclear what this sentence means.

Response: Below are the modified sentences. They can also be found in line 271-280

”The term symbiosis was introduced in the late 1870s after several studies on lichens had shown that an individual lichen consists of individuals of more than one species [6]. Symbiosis was then considered as a source of evolutionary innovation ([6] and [42] and their references). Symbiosis of prokaryotic cells is suggested to result in eukaryotic cells [3], which is considered one of the major evolutionary transition by Maynard Smith and Szathmary [43], and this is only one extreme case of obligate symbiosis in which obligate dependency is required from both partners. Examples of obligate and facultative symbioses are ubiquitous, with different levels of dependency from both partner. Yet, how obligate symbioses may evolve out of facultative ones is not well-understood, thus studying this intriguing transition process may give insights and guide for future studies. ”

# 409 - 411: The fact that Trichinella can be induced to grow on intestinal epithelial cells in a lab has no bearing on how obligate its symbiosis is in nature. Particularly as the intestine forms part of its natural habitat. What matters is what happens in nature.

Response: Meerovitch et al 1965 successfully cultured Trichinella Spiralis in axenic environment. The basal medium used inactivated normal rabbit serum and 25 % chick embryo extract [6].

# 414: sacrifice

Response: This has been fixed in line 328

# Figures, generally: Please provide labels A and B on all the plots, and please left-align, dont right-align these labels. I have never seen this done before and its very confusing. All the plots should have informative axis titles like Fig. 5 so they can more easily stand alone.

Response: The figures have been modified accordingly.

# Fig. 2: From the caption and figure alone, its unclear whether this trade-off is a result or an assumption. Obviously this is stated in the text, but it just doesn’t help comprehension to have this much ambiguity.

Response: The caption has been modified accordingly.

References

1. Poulin, R. Evolutionary Ecology of Parasites 2nd ed. (Princeton Univerity Press, 2006).

2. Miyokawa, R. Y. et al. Horizontal transmission of symbiotic green algae between hydra strains. Arch. Am. Art J. 235, 113–122. (2018).

3. Zook, D. in Reticul. Evol. Symbiogenesis, Lateral Gene Transf. Hybrid. Infect. Hered. 41–80 (2015).

4. Russell, S. L., Corbett-Detig, R. B. & Cavanaugh, C. M. Mixed trans- mission modes and dynamic genome evolution in an obligate animal- bacterial symbiosis. ISME J. 11, 1359–1371 (June 2017).

5. Bright, M. & Bulgheresi, S. A complex journey: transmission of microbial symbionts. Nat. Rev. Microbiol. 8, 218–230 (2010).

6. Meerovitch, E. Studies on the in Vitro Axenic Development of Trichinella Spiralis. Ii. Can. J. Zool. 43, 81–85 (1965).

---

## [Decision Letter · Decision Letter 1]

30 Mar 2020

PONE-D-19-25560R1

On the difficult evolutionary transition from free-living lifestyle to obligate symbiosis

PLOS ONE

Dear Mrs Nguyen,

Thank you for submitting your manuscript to PLOS ONE. After careful consideration, we feel that it has merit but does not fully meet PLOS ONE’s publication criteria as it currently stands. Therefore, we invite you to submit a revised version of the manuscript that addresses the points raised during the review process.

Thanks for your careful work on this and responding to the prior reviews. The current reviewer has some minor comments that I would like for you to attend to before publication. The article will not need to go out for review again, but there is no way in the PLOS system to allow you to modify the article if we push it straight to accepted.

We would appreciate receiving your revised manuscript by May 14 2020 11:59PM. To enhance the reproducibility of your results, we recommend that if applicable you deposit your laboratory protocols in protocols.io, where a protocol can be assigned its own identifier (DOI) such that it can be cited independently in the future. For instructions see: http://journals.plos.org/plosone/s/submission-guidelines#loc-laboratory-protocols

We look forward to receiving your revised manuscript.

Kind regards,

Stephen R Proulx

Academic Editor

PLOS ONE

Reviewers' comments:

Reviewer's Responses to Questions

**Comments to the Author**

1. If the authors have adequately addressed your comments raised in a previous round of review and you feel that this manuscript is now acceptable for publication, you may indicate that here to bypass the “Comments to the Author” section, enter your conflict of interest statement in the “Confidential to Editor” section, and submit your "Accept" recommendation.

Reviewer #2: (No Response)

2. Is the manuscript technically sound, and do the data support the conclusions?

Reviewer #2: Yes

3. Has the statistical analysis been performed appropriately and rigorously? 

Reviewer #2: N/A

4. Have the authors made all data underlying the findings in their manuscript fully available?

Reviewer #2: Yes

5. Is the manuscript presented in an intelligible fashion and written in standard English?

Reviewer #2: Yes

6. Review Comments to the Author

Reviewer #2: The authors investigate the evolution of obligate symbiosis when symbionts can exist on a continuum between completely free-living, facultative, and obligate. They model a trade-off between host encounter and reproduction, so that symbionts that are best at finding a host reproduce least, and may even require their host's help in order to reproduce at all. Interestingly, while symbionts in the model can be vertically transmitted, vertical transmission is both independent of obligacy and not sufficient to maintain a symbiont population. Instead, both facultative and obligate symbionts produce free-living progeny while in symbiosis.

This is an interesting model that seems applicable to many symbioses. The model of obligate symbionts that are not purely vertically transmitted seems like a great way to separate the evolution of obligacy from the evolution of transmission mode.

I only have some minor comments.

The authors mentioned in the supplement that they didn't determine the stability of the positive equilibrium when the trivial equilibrium was unstable. It would be nice add to the supplement some information about the stability of that equilibrium for a few parameter combinations, just as reassurance that nothing strange is going on for some reasonable parameter values.

Fig. 1 appears to be missing.

Lines 82-83: Since symbionts can always reproduce inside the host in this model, it might be better to say that free-living organisms are distinct from facultative symbionts because they are unable to enter into a symbiotic relationship (beta = 0).

Lines 109-110: I found this a bit vague. Would it be possible to add that the host population size is kept constant at N?

Fig 6 caption: "Black points indicate the ESS values that are corresponded with different values of B." I couldn't find any black points. Does this possibly refer to the colors in the plots?

Line 254: Is "condition (x)" supposed to have a number in place of x?

Lines 263-265: I think this should be "Then, b is no longer a constant but depends on x. As this involves consideration of how the host responds to the symbiont, and thus of coevolution..."

Line 301-303: I don't understand this sentence. Does it mean that x will be the same in both cases or that b will become 0?

Supplement: mu here is mu_0 in the main text.

7. PLOS authors have the option to publish the peer review history of their article (what does this mean?). If published, this will include your full peer review and any attached files.

Reviewer #2: No

---

## [Author Response · Author response to Decision Letter 1]

12 May 2020

Dear reviewer,

Thank you very much for reviewing our manuscripts. You will find below our response to each point that you raised, and we hope that it will clarify your questions.

\\# The authors mentioned in the supplement that they didn't determine the stability of the positive equilibrium when the trivial equilibrium was unstable. It would be nice add to the supplement some information about the stability of that equilibrium for a few parameter combinations, just as reassurance that nothing strange is going on for some reasonable parameter values.

Response: Indeed, we did not consider the stability of the equilibrium in the first submission. To show that this is not so important in our simple model, and that there is no strange behaviour of the equilibrium, we added in the supplementary document the bifurcation analysis of the equilibrium with respect to the independent reproduction rate $\\rho$ and the mortality rate of the association $\\nu$. It shows that if the reproduction is not sufficient or if the mortality rate is too high then the trivial equilibrium is stable, whereas in the reverse conditions, there are two stable non zero equilibrium, of which only one is positive and is feasible for the analysis.

\\# Fig. 1 appears to be missing.

Response: We changed the order of the figures when we edited the manuscript and some erroneous references escape our attention. Thank you for pointing this out. We corrected the error.

\\# Lines 82-83: Since symbionts can always reproduce inside the host in this model, it might be better to say that free-living organisms are distinct from facultative symbionts because they are unable to enter into a symbiotic relationship (beta = 0).

Response: We agree with the reviewer for this clarification and modified the phrase as followed. It is in line 80-83 in the revision manuscript.

"A complete loss of independent reproduction is equivalent to obligate symbiosis, whereas nonzero independent reproduction suggests facultative symbioses with different levels of independent reproduction. Purely free-living organisms do not form associations with the hosts even though they are available."

\\# Lines 109-110: I found this a bit vague. Would it be possible to add that the host population size is kept constant at N?

Response: We added the clarification in line 114-116 in the revision manuscript. Followed is the modified phrase.

"As in this study we aim to focus on the evolutionary pressures on the symbionts, we assume that the total host population size is a constant number $N$, and we will only consider host dynamics in a future study."

\\# Fig 6 caption: "Black points indicate the ESS values that are corresponded with different values of B." I couldn't find any black points. Does this possibly refer to the colors in the plots?

Response: This is an error that we made during the revision of our manuscript. This phrase should be in the caption of Figure 4, and we corrected this mistake.

\\# Line 254: Is "condition (x)" supposed to have a number in place of x?

Response: This is again an error that we failed to notice. Thank you for pointing this out. We changed the phrase into

"A consequence of this relaxation is that $\\rho'(x) = \\tau'(x)$ no longer holds, therefore $d \\tau/d \\rho$ is no longer a constant, which greatly simplified our analysis."

This is modified in line 255 in the main text.

\\# Lines 263-265: I think this should be "Then, b is no longer a constant but depends on x. As this involves consideration of how the host responds to the symbiont, and thus of coevolution..."

Response: We modified the phrase as followed. We hope that this will clarify what we meant.

"Then, b, the bonus reproduction gained from the exploiting the host, is no longer a constant but depend on x. Moreover, this involves consideration of how the host responds to the symbiont, and thus of coevolution. This falls outside of the scope of this article." 

It can be found in line 265-268 in the main text.

\\# Line 301-303: I don't understand this sentence. Does it mean that x will be the same in both cases or that b will become 0?

Response: We modified the paragraph so that our message is clearer. We basically meant that if the symbiont invests the same in independent reproduction and bound reproduction, i.e. $\\rho'(x) = \\tau'(x)$, evolution toward obligate symbiosis may be more difficult than if it invests differently in the two reproduction rats, i.e. $\\rho'(x) \\neq \\tau'(x)$. In the latter case, the evolution of losing independent reproduction may be favoured depending on the specific assumption of $\\rho'(x)$ and $\\tau'(x)$. 

The following modified paragraph can be found in line 297-306 in the main text. 

"This is because we assume that a symbiont invests in its free reproduction the same way that it does in its bound reproduction; what eventually keeps the symbiont in the association is the reproduction bonus that it receives from its host, which is assumed to be a constant amount in the scope of our model. 

There may be two reasons that the symbionts invest the same in reproduction regardless of whether it is free-living or associated with its host. First, the symbiont is not sensitive enough to perceive the differences between the external and the host environment. Second, the two environments are so similar that the symbiont does not need to respond differently. For either reason, obligate symbiosis is unlikely to evolve."

\\# Supplement: $\\mu$ here is $\\mu_0$ in the main text.

Response: We thank the reviewer for pointing this out. We changed the mathematical annotations in the supplementery document so that it matches those in the main text.

---

## [Editor Report · Decision Letter 2]

24 Jun 2020

On the difficult evolutionary transition from the free-living lifestyle to obligate symbiosis

PONE-D-19-25560R2

Dear Dr. Nguyen,

We’re pleased to inform you that your manuscript has been judged scientifically suitable for publication and will be formally accepted for publication once it meets all outstanding technical requirements.

Thank you for completing these changes. The addition of the stability analysis is a nice touch. I would like to apologize for the delay in processing your re-submission, it is entirely due to me being caught up with other responsibilities and I apologize for the long wait.

Kind regards,

Stephen R Proulx

Academic Editor

PLOS ONE
---

## [Editor Report · Acceptance letter]

6 Jul 2020

PONE-D-19-25560R2 

On the difficult evolutionary transition from the free-living lifestyle to obligate symbiosis 

Dear Dr. Nguyen:

I'm pleased to inform you that your manuscript has been deemed suitable for publication in PLOS ONE. Congratulations! Your manuscript is now with our production department. 

Kind regards, 

on behalf of

Dr. Stephen R Proulx 

Academic Editor

PLOS ONE